# Physically Acting Treatments for Head Lice—Can We Still Claim They Are ‘Resistance Proof’?

**DOI:** 10.3390/pharmaceutics14112430

**Published:** 2022-11-10

**Authors:** Ian F. Burgess

**Affiliations:** Medical Entomology Centre, Insect Research & Development Limited, 6 Quy Court, Colliers Lane, Stow-cum-Quy, Cambridge CB25 9AU, UK; ian@insectresearch.com; Tel.: +44-1223-810070

**Keywords:** clinical trials, essential oils, head lice, insecticides, resistance, silicones, surfactants, treatment

## Abstract

Head lice worldwide have developed resistance to insecticides, prompting the introduction of a range of alternative treatments including plant extracts and natural and synthetic oils. Clinical studies of physically acting treatments showed them to be highly effective when first introduced, and a widely held, but unsubstantiated, belief is that lice are unlikely to develop resistance to them. However, this ignores possibilities for natural selection of traits enabling lice to survive exposure. More recent investigations of some physically acting products have shown reduced efficacy, suggesting either changes of behavior, physical structure, or physiology of some louse populations. In addition, the activity of surfactants and similar compounds, acting as solubilizing agents of insect cuticular lipids, can be compromised by the widespread use of toiletry products containing similar substances. Hitherto, most clinical investigations have provided “best case” data resulting from investigator application of treatments. In the few studies involving participant application, the effectiveness was reduced, suggesting that consumer use allows some insects to survive, which could then be selected for tolerance. Unlike neurotoxic insecticides, there is no straightforward method to test for the activity of physically acting chemicals other than by clinical investigations, which need to be rigorous to eliminate poorly effective products as a way of ensuring the continued effectiveness of those treatments that are successful in eliminating infestation.

## 1. Introduction

Resistance to treatments for human louse infestation has arisen throughout the history of attempts to eradicate lice by chemical means. Any pharmacologically acting insecticide that inhibits an enzyme or disrupts a physiological process is inevitably susceptible to detoxification itself, in much the same way as any environmental toxicant is countered by the physiology of the insect. The only difference is that insecticides are intended to be used at levels that aim to kill the insect before it can denature the insecticide. However, inadequate dosing or vigor tolerance permitted some insects to survive the encounter and from that cohort of the population resistant insects were selected, either because they were more efficient at producing the necessary enzymes or else because a mutation was selected that altered their physiology to the extent that the insecticide no longer exerted its effect [1].

Acquired resistance to pyrethroid insecticides in head lice, based on point mutations of the α-subunit of the voltage-gated sodium channel (*kdr* resistance), previously identified in houseflies following prolonged dichlorodiphenyltrichloroethane (DDT) use [2], arose in the early 1990s and has since become worldwide in distribution [3,4,5,6,7,8,9,10,11,12] in parallel with loss of sensitivity to other insecticides [13,14,15]. Despite repeated reports of treatment failure, most pediculicide manufacturers failed to take positive action to address the problems, relying on historical data [16], professional assessments related to the thoroughness of treatment [17], or reports of better efficacy of specific products from some geographic regions [18].

One approach to address the problems arising from resistance was a consumer revival of interest in plant extracts, essential oils, and other herbal components. Apart from a somewhat naïve belief that these materials are inherently safer than all synthetic compounds, it was also claimed that the chemical complexity of most plant extracts makes them less likely to be affected by resistance, and their natural origin to minimize their environmental impact. Scientific evidence for many natural products is limited [19,20,21,22].

Claims are routinely made that plant derived products act against insects by physical rather than physiological mechanisms. This can reasonably apply to fixed plant oils, such as unrefined coconut oil or pressed olive oil, but most plant extracts, especially essential oils, operate through physiological mechanisms. For example, tea tree oil, a popular essential oil distillate from *Melaleuca alternifolia* used against lice [23], as with most mixtures of monoterpenes and terpene esters, can exert a solvent effect on cuticular lipids, but the most active components: terpinen-4-ol, *trans*-anethole, and 1,8-cineol are all neurotoxins acting as acetylcholinesterase inhibitors in a similar manner to the mode of action of organophosphorus insecticides [24,25,26,27,28,29,30]. Therefore, contrary to the perception that natural materials are not affected by resistance, it is likely that a cross-resistance occurs between essential oils and synthetic insecticides, rendering the natural materials ineffective and enhancing insecticide resistance by selecting insects capable of detoxifying both groups of chemicals.

The risks of cross-resistance between essential oils and neurotoxic insecticides, or different mechanisms developing in parallel that can affect essential oils as well as insecticides, have been mentioned above. As an example, one formulation with 0.5% malathion in an alcoholic basis also contained approximately 12% of the essential oil terpenoids *d*-limonene and α-terpineol. This product was reformulated after about 20 years to remove the terpenoids to improve the odor. Within a few weeks, complaints of failure were reported and it was later confirmed that the new preparation was less effective because much of the original activity was conferred by the terpenoids [31]. In 1999 the original alcoholic malathion formulation with terpenoids was used as a rescue treatment in a clinical study [32]. Although some lice in the area were malathion resistant it was assumed that the 12% terpenoids would be curative but some lice could be repeatedly immersed in the fluid and survive. Lice treated using two different malathion preparations in the laboratory gave similar results (Table 1). For the aqueous emulsion (Derbac-M liquid, Reckitt Benckiser Group, plc, Slough, UK), the resistance was to malathion alone. Failure of the alcoholic product with terpenoids (Prioderm lotion, Napp Laboratories, Cambridge, UK) indicated acquired resistance to both malathion and the terpenoids, with lice not affected by either the neurotoxic activity or any solvent activity of the terpenoids on cuticle lipids.

Consequently, only truly physically acting treatments could be considered unaffected by physiological resistance mechanisms. The first breakthrough alternative therapeutic agents not relying on physiological mechanisms were based on siloxanes (silicone) and proved highly effective in clinical studies in areas where insecticide resistance was prevalent. Overall, they were significantly more effective than most conventional insecticides [33,34,35,36], and not affected by the physiological resistance mechanisms affecting insecticides. Further studies demonstrated a physical mode of action by coating the insects and smothering them, putatively killing by “asphyxiation” [37], prevention of water excretion [38], or else by disrupting cuticle lipid integrity by dissolution or emulsification resulting in dehydration [32,39,40,41].

## 2. Current Trends

After more than 17 years of using siloxanes and other lipids and lipid emulsifying chemical products [42], how effective are these products in clinical use? If the hypothesis were to hold true that the physical mode of action operates in such a way that resistance is unlikely [43], or even not possible [44], then all the products should remain as efficacious as they were when first introduced. However, to eliminate entirely the possibility that lice could acquire a tolerance for physically acting treatments is at best naïve and at worst risky. It is naïve in that it eliminates from thought the possibility that insects may be selected by these products for different physical and/or physiological characteristics that could allow them to tolerate or avoid the active effects. Risky in that, as was seen with continued reliance on neurotoxic insecticides long past the point when they became ineffective, there could be a widespread failure to control the parasites that is either overlooked by complacency or dismissed due to inadequate treatment procedures [17].

### 2.1. Fixed and Synthetic Oils

Non-volatile oils with no pharmacological activity, glycerides of saturated and non-saturated fatty acids, and various mineral oils potentially have an occlusive effect on lice. The ability to thoroughly coat the louse surface depends upon the viscosity, surface tension, and wetting angle of the fluid, but the ability of the oil to maintain contact with the louse depends upon the flow characteristics of the fluid and the van der Waals forces applying to the interaction between the oil and the substrate, in this case the louse cuticle [45]. These oily materials can physically block the openings of the louse respiratory system, but in clinical use do not penetrate further than the spiracular gland constriction proximal to the spiracle [38]. This does not result in asphyxiation [46,47], and disruption of water excretion appears to be the primary effect [38,47]. Consequently, it is widely believed, and promoted by pharmaceutical manufacturers [43,44], that siloxanes, synthetic oils, and fatty acid ester-based products are effectively “resistance proofed”, primarily because there is no predictable pathway that would permit lice to develop resistance to this type of technology.

Not surprisingly, a certain “failure to cure” rate occurred in all but one of the clinical studies so far published [32,33,34,35,36,40,41,48,49,50,51,52,53,54,55,56,57,58,59]. The interpretation of failure has varied from study to study but, in all cases, there were some participants who had lice that unequivocally failed to respond to treatment. If failure to apply the product correctly, in sufficient quantity to cover the head/hair, and other compliance issues are eliminated, the question arises as to how lice could have or develop a tolerance to these treatments.

Experience with neurotoxic insecticides shows that resistance to those chemicals is primarily physiological. However, this resistance can be broken down into metabolic processes, target site insensitivity, penetration resistance, as well as behavioral avoidance effects [1]. In any population of organisms, including insects, there are some physical variations of structure, and for head lice the target site for oil treatments is the spiracle, which varies greatly between species [60]. The occlusion of spiracle structures depends upon several characteristics of the anatomy, such as the angle and size of the peritreme, the width of the spiracle aperture, and the complexity of the honeycomb-like structure within the atrium, apart from the flow and volatility characteristics of the applied fluid. Some changes of response by lice to immersion in water have been reported [47], with head lice continuing to move about, and in some cases crawl out from the liquid, where in the past they would have been immobilized within seconds, which suggests that either there is variation in the spiracle structure of individual lice or else the feedback mechanism that triggers the immobilization is less sensitive in some insects.

If lice can acquire tolerance of aggressive terpenoid solvents, other solvents and emulsifiers used to disrupt louse cuticular lipids, or even occlusive fluids, can be similarly affected by acquired tolerance, potentially rendering numerous new products ineffective. As an example, the isopropyl myristate/cyclomethicone (IPM/C) 50:50 mixture (Full Marks solution, Reckitt Benckiser Group, plc, Slough, UK) showed a reduction of efficacy between studies conducted a few years apart [35,48]. In 2005/6 the cure rate was 77% in two related trials, with 27/35 and 57/74 participants cured [35]. Approximately 18 months later we found a lower rate of 68%, with 36/53 participants louse free after two treatments (Odds Ratio 1.5867) [48]. However, since then repeated individual cases have reported failure to eliminate infestation, and recently a different IPM-based preparation showed a clinical success rate of only 41% (unpublished observations).

Was this difference between studies part of a continuing process of loss of effectiveness or the type of random observational variation that could occur in comparing any two clinical studies? If it was a loss of activity, would it be confined to just that product or active material or would it affect other and different formulations and dosage forms in the same way as tolerance of pesticides? Some products use isopropyl myristate (IPM) as a named active substance for use against lice, while others use cyclomethicone as a silicone solvent. One that does name IPM as an active substance in Europe is a pressurized mousse with isopropanol and water (Vamousse™, Alliance Healthcare Ltd., Chippenham, UK) for which online consumer reviews suggest there is considerable geographic variability in success [61]. However, ex vivo tests following the pack instructions performed on freshly collected UK head lice (Table 2) suggested that the bioavailability of the active material is insufficient to kill head lice even 18 h after treatment, which in consumer terms means lice would be exposed to the active IPM and survive the exposure, potentially leading to tolerance.

### 2.2. Surfactants and Similar Compounds

If physically acting products with differences of formulation, but containing the same named materials, demonstrate considerably different effectiveness, in much the same way as has been observed with pesticide formulations [31], how many products may not be as effective as claimed? Many of the products marketed in Europe, like neem shampoo and some of the silicone-based products [36,49,50,62,63], have only been evaluated in developing countries where lice may never have encountered many of the product excipients that are supposedly inactive. These excipient chemicals possibly have no effect on lice in countries with developed economies because they are included in so many other hair care preparations at low concentrations, so the lice have acquired a tolerance for them, but, when applied to chemically naïve lice, they can exert just as much killing activity as the named “active” substance. A good example illustrating this was a case report of treating permethrin resistant lice cleared from all subjects using only ethanol 96% in a community where use of short chain alkanols such as ethanol and isopropanol would not normally be considered for treating the head [64]. The converse was shown by the investigation of the surfactant cocamide diethanolamine (cocamide DEA), which was found effective in killing laboratory reared, and surfactant naïve, clothing lice (*Pediculus humanus humanus*) and their eggs with a single application [32]. However, when the product was used in community-based randomized clinical studies in the UK, where this chemical was often included in toiletry shampoos, it failed to achieve a cure rate better than 33.9% (19/56), irrespective of the dosing regimen used [32].

Even surfactants widely used as toiletries can show effectiveness in stripping the lipids from the louse surface if concentrated enough and left in contact for a long enough application time. An example of this was a study using a shampoo that claimed to contain soya oil [65], which was subsequently found to contain no soya but instead had sufficient sodium laureth sulfate that when left in situ for repeated 30-min applications was able to eliminate lice from 28/45 (62.2%) participants [65]. In this case it appears that the soya oil name was used as a “mask” to make the product appear more appealing to consumers wishing to employ natural materials for therapeutic purposes. Another shampoo from the same source with fractionated coconut oil as the named active substance also used high levels of surfactant, but it proved to be clinically effective in only 12/31 (38.7%) participants [66]. Using such a shampoo in a population not previously exposed to powerful surfactants could itself eliminate lice and may have influenced the outcomes of 1% permethrin creme rinse studies [67], in which the original green Prell (Procter and Gamble, Cincinnati, USA) “stripping” shampoo containing cocamide was used as a prewash before application of the insecticide in trials among the Guna people in the San Blas archipelago of Panama [68].

Some novel surfactants have been investigated for their ability to kill lice by the disruption of cuticle lipids. Several vicinal diols were shown to exert pediculicidal activity in laboratory screening, of which the most effective overall was 5% 1,2-octanediol because it was found to inhibit louse egg production [69]. In clinical investigations, this compound was most effective in an alcohol vehicle, but was also active in an aqueous basis if applied for 8 h or overnight [41]. The compound was found to disrupt cuticular lipids, removing three major hydrocarbons: C25, C27, and C29, plus various low concentration analytes from the louse surface, resulting in a loss of viability and ultimately death by dehydration ([41] supplementary data). Although it was significantly (*p* = 0.0129) better than placebo, this effect was less noticeable in a study using the same compound at 1% in a protective leave in product and had the disadvantage that many lice came into contact with the compound and survived the encounter [70]. Did some lice survive because their cuticular lipid make up differed from those that showed susceptibility? If so, some populations of lice exposed to 1,2-octanediol would be selected for tolerance, potentially with a stimulatory effect to produce different lipid combinations for waterproofing the cuticle. However, an additional action of diols is the inhibition of N-methyl-D-aspartate (NMDA) receptors [71], essentially a neurotoxic effect similar in outcome to inhibition of these receptors by ketamine, with the greatest activity against NMDA receptors by alcohols with a chain length shorter than C10 at saturating concentrations [72]. As with any neuroactive compound, 1,2-octanediol would therefore be subject to metabolic activity and physiological interactions to counter its effect on the louse nervous system in surviving lice, either through enzymatic action or enhanced excretion leading to tolerance or physiological resistance.

## 3. Testing for Resistance

Testing insects for susceptibility to unformulated chemicals and formulations containing pharmacologically active chemical substances like neurotoxic insecticides is relatively straightforward, with numerous published protocols and guidelines from the World Health Organization (WHO) and others [73,74,75]. In contrast, the nature of physically acting materials that rely on occlusive effects or lipid disruption make testing in vitro almost incapable of mimicking the relatively low level of contact with lice that occurs as these fluids are dispersed over and along hair shafts.

In the laboratory, one approach to testing involves the complete immersion of the insects in the fluid for a relatively prolonged period without draining off excess fluid, ensuring that any surface interaction to disrupt cuticular lipids or flow into the spiracles proceeds as completely as possible. In contrast, when applied to a head, immersion of the lice is momentary followed by a “draining” effect as the fluid spreads out over the hair and scalp so the surface of the insect may only retain a thin film of fluid, if any at all, depending upon the wetting characteristics of the product.

The alternative testing approach uses a standardized exposure time irrespective of how the product is intended for use, e.g., the now discontinued American Society for Testing and Materials (ASTM) standards for evaluating pediculicides and ovicides, which used body/clothing lice rather than head lice and always exposed the lice for 10 min even for products designed to be applied for several hours [74,75].

As indicated above, all laboratory tests of formulated products, irrespective of method, are inclined to give a “false positive” representation of efficacy because the level of contact between the insect and the preparation in the Petri dish is more thorough than can be achieved when a product is dispersed throughout a head of hair. Consequently, true assessments of whether they are effective can only be made clinically. In Europe the physically acting preparations are classed as medical devices and the majority have been regulated under the terms of the Council Directive 93/42/EEC of 14 June 1993: Article 1, Section 2, sub-section (k) and Annex X for clinical evaluation. How that should be done is also open to some question since the Directive also permits the demonstration of equivalence through the literature with an already approved device, or in vitro data only, so a manufacturer’s claims of “clinically proven” or similar words, e.g., for Lyclear/Paranix Treatment Shampoo (Perrigo Company plc, Dublin, Ireland) [76], but without reference to a published study, give no clarity of whether the product is actually effective. Historically, many products have been assessed in efficacy studies, i.e., trials in which the treatments were applied by members of the investigation team [3,18,22,23,32,33,34,35,36,40,47,48,49,50,51,52,53,54,56,57,58,59,62,64,65,66,67,69], which ensures a thorough dosage and coverage, so the result is a potentially best possible outcome. Few published studies describe pragmatic or effectiveness studies [40,55,63,70] in which the treatment was given to the care giver and applied by them rather than by an investigator. Some studies in this model experienced a high rate of exclusion from analysis from non-compliance or drop out [55,70], so it was difficult to identify the true effectiveness of the applied product. All but one [52] study, have resulted in some level of failure to cure, and very few gave a cure rate close to the ideal proposed by Vander Stichele and colleagues [16], who stated “*Moreover, inspection of the figure* [not shown here] *leads us to recommend that only products with an expected cure rate of over 90% should be tested and that this should be done in trials with sufficient power to establish cure rates with a lower confidence limit above 90%*”. How such a high cure rate could be reliably predicted is impossible to determine because in vitro/ex vivo tests are mostly poor indicators of efficacy and cannot address human and environmental factors likely to influence clinical outcomes.

## 4. Discussion

Recent clinical observations and consumer reports both suggest that at least some of the physically acting preparations are losing effectiveness. Just as with the neurotoxic insecticides in the 1980s and 1990s, this phenomenon is occurring slowly with as yet no substantiation and can easily be written off as failure by care givers to adequately apply the treatment [17]. This is certainly a factor contributing to the effect, partly because some people have become blasé about the claimed efficacy of products, partly because they are trying to economize when faced with repeated need to treat, and partly through lack of skill. However, those same factors contributed to the development of acquired resistance to commonly used insecticides like permethrin and malathion and, when the warning signs of consumer dissatisfaction 25–30 years ago were not heeded, it resulted in complete loss of usefulness of the insecticides in most territories and regions within a little more than a decade [3,4,5,6,7,8,9,10,11,12,13,14,15,77,78,79].

Irrespective of how well products might perform when applied by investigators, how well would they work when applied by consumers? We have seen an interesting metamorphosis of reporting since the introduction of these products in about 2005, from complete satisfaction and more or less every time cure through to repeated treatment failures. Of course, some products that are reported as failing either have never been subjected to clinical investigations or else such investigations have never been published, but even those products that may have given acceptable results in clinical investigations have been reported as failing repeatedly by caregivers. In some cases, this is definitely due to inadequate application of the treatment, but some appear to be due to survival of either lice or their eggs after having been thoroughly saturated.

As already mentioned, identifying a mechanism whereby lice could now survive a treatment to which their ancestors were wholly susceptible is quite difficult, especially if the perceived mode of action is one of occlusion of some anatomical feature such as the blocking of spiracle openings. As shown by histology [60], the spiracles of lice are sclerotized, rigid objects inhibiting the ingress of fluids into the respiratory tract. Such structures do not change easily in response to selection pressures, unlike degradative enzyme responses to neurotoxic insecticides. Incorrectly applied treatments would exert no physiological stimulus and no “mutational” effect to change the spiracle structure. But lice that happen to have physically smaller spiracles or that are structured differently may be less susceptible to fluid entry. These features could be heritable traits, so their offspring could be more successful at surviving a more intense exposure to the same treatment.

When siloxane products were first used, lice were reported to run away from the advancing low viscosity fluid as it spread along the hairs [33]. Over time this behavioural trait has disappeared, suggesting that if the preparation is insufficiently viscous to result in a build-up on the louse cuticle the insects are more tolerant of its presence. In the failed isopropyl myristate study referred to above, subjects were treated on Days 1 and 8, with combing on Days 2, 7, 8, 10, and 15 to check for lice. Table 3 shows the number of lice, which appeared to be able to feed and reproduce normally, that were combed out from those subjects whose treatment was ineffective. Some lice combed from treated heads appeared to tolerate it without ill effects (unpublished observations). These lice were observed to carry a film of the treatment fluid on their surfaces but were only killed if completely immersed in the fluid (Table 4), something that could not physically occur on a patient’s head.

The introduction of surfactant-based treatments theoretically attacked lice using a mode of action unrelated to that of oily fluids. Therefore, the aim of introducing lipid disrupting surfactants, apart from providing consumers with alternative types of treatments, would have been to mitigate any risk of applying excessive selection pressure to the occlusive liquids. However, apart from any problems with occlusion, the siloxanes and other oily materials also disrupt cuticular lipids in a similar manner to surfactants. A comparison of the gas chromatograms of cuticular lipids removed by isopropyl myristate/cyclomethicone (IPM/C) and by 5% 1,2-octanediol shows that both formulations remove the same lipids with C25, C27, and C29 carbon chains [39,41], the principal lipids on the cuticle, found to form about 46% of the total cuticular lipid in head lice [80]. This means that any perceived benefit obtained by using different groups of chemicals with physical activity on the external characteristics of lice may be spurious in terms of avoiding loss of sensitivity and may exacerbate any risk of lice acquiring resistance to current physically acting treatments. We have already seen the loss of effect of surfactants such as cocamide DEA [32] and oils like IPM/C ([48] unpublished observations), so it is only a matter of time before loss of activity from other compounds is reported. If head lice are being selected for greater tolerance of being soaked in oily fluids and surfactants, action needs to be taken now by pediculicide manufacturers, clinical investigators, and regulators to establish which types of formulations and which “active” materials are losing their effect.

What, therefore, are the greatest potential drivers of acquired resistance to physically acting treatments? Much as with neurotoxic insecticides, simple factors such as adequate and thorough application and leaving products in situ for long enough periods are fundamental to successful treatments [1,17]. However, recent marketing trends seem to ignore these factors and, in attempts to take maximum market shares, various companies appear to be involved in a “race to the bottom” regarding dosing and application time. As examples, when first introduced, dimeticone-based products like Hedrin 4% solution (Thornton & Ross Ltd., Huddersfield, UK) and Nyda L (Pohl-Boskamp GmbH & Co. KG, Hohenlockstedt, Germany) were applied for 8 h or overnight [33,34,36] but following an in vitro study that showed 100% efficacy after 5 min [81], the application time of Nyda has been widely reduced to 10 min [82], and in some territories, such as Spain, to 5 min [83]. Similarly, the application time of Hedrin/Neositrin Spray Gel (Stada Group, Bad Vilbel, Germany) has been reduced in Spain from the original 15 min [52] to just 1 min [84] and a similar pattern has emerged across the industry in countries where competition for market share is intense and claims for shorter application times or more treatments from a pack may increase consumer purchasing. Inevitably reducing application time and spreading the treatment more thinly increases the chance that lice may encounter the product without being killed. As with neurotoxic insecticides, it is from that cohort of survivors that resistant insects can be selected.

## 5. Conclusions

Physically acting treatments for head lice have made a considerable positive impact on the control and management of insecticide resistant populations of lice. So much so that they have virtually ousted insecticide-based products from the market in several European and other countries. However, the interpretation of the term “physically acting” is somewhat loose in some regulatory jurisdictions, so some of the products making this claim may be just as sensitive to the risks of acquired resistance as the insecticides that preceded them because there is alternative evidence that the chemicals in questions have a physiological activity in addition to any physical effect.

Irrespective of the physical nature of the activity of a preparation, the idea that lice cannot become resistant to it is a false concept. Insects have demonstrated an ability to develop resistance to a wide range of killing measures over the past 100 years, and there is no reason to justify a belief that synthetic oils and other physically acting chemicals are exempt from this risk. As with claims about fixed vegetable oils like coconut, which may kill lice in urban communities in developed countries, these are only good for use against lice that have never encountered them, unlike the lice in Africa and Asia where the oil is widely used as a hair conditioning treatment. Regular use of these materials can result in sub-lethal encounters that may select for lice able to tolerate the exposure because they have some difference of physical characteristics or physiology, and the same may apply to physically acting chemicals just as much. Consequently, in order to avoid problems in the future for the currently successful products, a greater degree of care and thoroughness is required in their pre-marketing evaluation, marketing, and instructions for consumer use.

**Literature search**: Databases searched include PubMed, Science Direct, Scopus, Cochrane, Google Scholar, https://worldwide.espacenet.com, http://phthiraptera.myspecies.info (accessed on 13 June 2022), as well as hand searching online, https://clinicaltrials.gov (accessed on 13 June 2022), the ISRCTN registry (https://www.isrctn.com/ (accessed on 13 June 2022)), and my own collection of more than three thousand physical and electronic reprints and references, using terms including: head lice, pediculosis, treatment, clinical trials, and more specific target terms such as “physically acting treatment” or, “non-insecticide treatment”.

## Figures and Tables

**Table 1 pharmaceutics-14-02430-t001:** Survival of resistant head lice treated using different 0.5% malathion preparations, one a simple aqueous emulsion and the other an alcoholic solution containing 12% terpenoids.

Malathion Product	Number of Lice			Mortality %
	**Total**	**Alive**	**Killed**	
Aqueous emulsion	89	47	42	47.2
Alcohol + terpenoid	94	63	31	33.0
Control (water)	42	40	2	4.8

**Table 2 pharmaceutics-14-02430-t002:** Effect of Vamousse™ on head lice recorded 18 h after treatment.

Treatment	Number of Lice				Mortality %
	**Total**	**Alive**	**Moribund**	**Killed**	
Vamousse™	15	10	3	2	33.3
Control (water)	21	19	1	1	9.5

**Table 3 pharmaceutics-14-02430-t003:** Viable lice recovered following treatment with an isopropyl myristate product.

Number of Subjects	Days after Treatment Day (Day 0)				
	**Day 2**	**Day 7**	**Day 8**	**Day 10**	**Day 15**
With lice (no. lice)	11 (24)	12 (63)	9 (101)	9 (39)	10 (53)
Without lice	11	9	7	11	12

**Table 4 pharmaceutics-14-02430-t004:** Effect of in vitro immersion of lice in the isopropyl myristate product.

Treatment	Number of Lice				Mortality %
	**Total**	**Alive**	**Moribund**	**Killed**	
IPM product	52	0	0	52	100
Control (water)	20	19	0	1	5.0

## Data Availability

The data presented in this study are available on request from the corresponding author. The data are not publicly available because some have not yet been released by a sponsor.

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
