# Peer review of "Physically Acting Treatments for Head Lice—Can We Still Claim They Are ‘Resistance Proof’?"

_pharmaceutics, 2022, doi:10.3390/pharmaceutics14112430_

Round 1
Reviewer 1 Report
The manuscript covers an important question and various factors are discussed.
Author Response
Reviewer has made no specific comments but may wish to see the revisions proposed by other Reviewers
Reviewer 2 Report
This review from Prof. Burgess is very timely, well written and important to the field. I have no ammendments to this manuscript.
Author Response
Reviewer has made no points that require response but may wish to see changes prompted by other Reviewer comments
Reviewer 3 Report
The manuscript redacted by Ian F. Burgess is a critical review focused on different approaches for treating head lice, alerting the need for new effective options in the near future due to their resistance to those currently available. The manuscript is well redacted, showing the author's point of view and concern about the topic.
Also, the author sharply alert to the way assays are performed in order to address this activity, redacting that “all laboratory tests (…) are inclined to give a “false positive” representation of efficacy”.
Q1: In the case of publishing this manuscript in the present form, is the author properly aware of how this could be problematic? Is the real intention of the author to keep this sentence?
Probably because author's experience concerning this topic, something evident is the extensive argument that is not supported by literature, I mean, without references. In this regard, it would be great if adding literature to the following: lines 29-39, 78-89, 113-116, 124-127, 244-256.
Line 42: DDT is presented as an acronym; it should be first introduced. Please add the DDT meaning in the text.
Line 77: “After more than 17 years (…)”, this really needs a reference.
Line 99: “These oily (…)”, if this sentence is related to the previous paragraph, it should not be a new one.
Line 128: “I have already (…)”, it would be great to avoid personal statements. Please rephrase this sentence.
Line 136: “In 1999 (…)”, which original formulations? The same as for line 99, if this sentence is connected with the previous, this should not be a new paragraph.
Line 208: “in situ” in italic form, please.
Line 316-324: Data about the info discussed should be presented to this manuscript to support the paragraph, otherwise this paragraph should not be published from my point of view. As the author provides assays in tables 1 and 2, this topic should obey to the same criteria.
Author Response
Also, the author sharply alert to the way assays are performed in order to address this activity, redacting that “all laboratory tests (…) are inclined to give a “false positive” representation of efficacy”.
Q1: In the case of publishing this manuscript in the present form, is the author properly aware of how this could be problematic? Is the real intention of the author to keep this sentence?
Response: This sentence has been redrafted, including the original statement but with a clarifying qualification as follows, "As indicated above, all laboratory tests of formulated products, irrespective of method, are inclined to give a “false positive” representation of efficacy because the level of contact between the insect and the preparation "in the Petri dish” is more thorough than can be achieved when a product is dispersed throughout a head of hair"
Probably because author's experience concerning this topic, something evident is the extensive argument that is not supported by literature, I mean, without references. In this regard, it would be great if adding literature to the following: lines 29-39, 78-89, 113-116, 124-127, 244-256.
Response: References have been inserted in the places indicated. For lines 29-39 and new reference "0" has been inserted; lines 78-89 (lines 102-104 and 126 in the revised version) references 40a-40c ; lines 113-116 lines reference "0" has been cited again; 124-127 (now 149-152) have been deleted because there are no referenced observations; lines 245-256 additional references 71-73 have been inserted.
Line 42: DDT is presented as an acronym; it should be first introduced. Please add the DDT meaning in the text.
Response: The full name "dichlorodiphenyltrichloroethane" has now been entered in the text with "DDT" in parentheses.
Line 77: “After more than 17 years (…)”, this really needs a reference.
Response: A new reference (40a) has now been entered that shows this type of product was introduced in 2005.
Line 99: “These oily (…)”, if this sentence is related to the previous paragraph, it should not be a new one.
Response: The two paragraphs have been joined to form a single paragraph.
Line 128: “I have already (…)”, it would be great to avoid personal statements. Please rephrase this sentence.
Response: This sentence has now been rephrased.
Line 136: “In 1999 (…)”, which original formulations? The same as for line 99, if this sentence is connected with the previous, this should not be a new paragraph.
Response: The identification of the formulation has now been clarified by changing the description to "the original alcoholic malathion formulation with terpenoids" and the two paragraphs have been joined back together as a single paragraph.
Line 208: “in situ” in italic form, please.
Response: This has now been done.
Line 316-324: Data about the info discussed should be presented to this manuscript to support the paragraph, otherwise this paragraph should not be published from my point of view. As the author provides assays in tables 1 and 2, this topic should obey to the same criteria.
Response: this paragraph has been partially reworded and two tables of data (table 3 and Table 4) have been inserted to explain this text.
Reviewer 4 Report
It is obvious that a lot of thought has been put into this review and it is certainly publish worthy. The subject is important and is clearly an area that the author is competent in. The paper could be tightened up a bit as it tends to waiver. I think there is a need for this review. One issue you point out is pressure from manufacturing companies to push false narratives about their products. Good luck at fixing that. You repeat often the "widely known belief" that physical treatments for insects are immune from resistance, but you do not provide specific references to support this current dogma. You need to add specific references for this widely held belief. I for one do not accept this belief. Physical treatments, just as for most all other attempts to eradicate a pest, can result in a heritable change in sensitivity of the pest population that leads to a declining efficacy of a product. And as you state there are many barriers to proper evaluations in field trials and resultant "failures" of a product. Rigorous clinical trials are needed as you state. But again you fail to bring forward specific references that are causing your heartburn on the subject. Where are the pressures coming from to make you so outspoken on this idea. Is this also widely known? Provide specific references to support these statements. When you reference a product name or company along these lines you still need to give a specific example (in the referencing) where these pharmaceutical companies are failing to understand evolution and adaptations in insects. One area that maybe could be discussed a bit is behavioral resistance to a product by an insect which could include changes of feeding behavior or escape behavior. Many examples of mosquitoes behavior that leads to a pest population avoiding a treatment area are in literature. Cockroaches developed a resistance to some baits based only on the sugar attractant in the bait. This change in behavior can have dramatic affects on the efficacy of a physical acting product as well.
Author Response
The paper could be tightened up a bit as it tends to waiver.
Response: The manuscript has been checked over and one section moved to make it more consistent in the train of thought. Additional statements have been made regarding clinical investigations and the evidence for efficacy between lines 265 and 277. Other smaller changes have been made to tighten it up and a final paragraph about possible drivers for resistance added to the Discussion to qualify the point.
You repeat often the "widely known belief" that physical treatments for insects are immune from resistance, but you do not provide specific references to support this current dogma. You need to add specific references for this widely held belief.
Response: Some references (40b and 40c) have been added as requested.
And as you state there are many barriers to proper evaluations in field trials and resultant "failures" of a product. Rigorous clinical trials are needed as you state. But again you fail to bring forward specific references that are causing your heartburn on the subject. Where are the pressures coming from to make you so outspoken on this idea. Is this also widely known? Provide specific references to support these statements.
Response: This point is a more difficult one to address fully. I have mentioned in the text, and referenced, some of the problems associated with conducting trials of products designed for use in developed economies in countries with developing economies where people/lice have not used most of the chemical entities found in the products under test. There is also an issue with the regulatory environment with enforcement of the Medical Devices Directive requirements for clinical data and a short section addressing that has been added (lines 270-277).
One area that maybe could be discussed a bit is behavioral resistance to a product by an insect which could include changes of feeding behavior or escape behavior.
Response: This also is a difficult area to address other than superficially. One instance has been mentioned in relation to lice "running away" from early forms of silicone fluids but no longer doing that (lines 328-331) . Unfortunately, our general level of knowledge on details of louse behavior falls far below that of free living insects like mosquitoes and cockroaches so, much as I would like to be able to provide detailed comments and references to address this point, I regret the information is just not there.